# Analysis of Shape Memory Behavior and Mechanical Properties of Shape Memory Polymer Composites Using Thermal Conductive Fillers

**DOI:** 10.3390/mi12091107

**Published:** 2021-09-15

**Authors:** Mijeong Kim, Seongeun Jang, Sungwoong Choi, Junghoon Yang, Jungpil Kim, Duyoung Choi

**Affiliations:** 1Carbon & Light Material Application Research Group, Korea Institute of Industrial Technology, Jeonju 54853, Korea; kmj03007@kitech.re.kr (M.K.); jse95@kitech.re.kr (S.J.); chl6034@kitech.re.kr (S.C.); jyang@kitech.re.kr (J.Y.); jpkim@kitech.re.kr (J.K.); 2Division of Mechanical Design Engineering, Jeonbuk National University, Jeonju 54896, Korea

**Keywords:** shape memory polymer, thermal conductive filler, shape recovery rate, thermal conductivity, curing condition

## Abstract

Shape memory polymers (SMPs) are attracting attention for their use in wearable displays and biomedical materials due to their good biocompatibility and excellent moldability. SMPs also have the advantage of being lightweight with excellent shape recovery due to their low density. However, they have not yet been applied to a wide range of engineering fields because of their inferior physical properties as compared to those of shape memory alloys (SMAs). In this study, we attempt to find optimized shape memory polymer composites. We also investigate the shape memory performance and physical properties according to the filler type and amount of hardener. The shape memory composite was manufactured by adding nanocarbon materials of graphite and non-carbon additives of Cu. The shape-recovery mechanism was compared, according to the type and content of the filler. The shape fixation and recovery properties were analyzed, and the physical properties of the shape recovery composite were obtained through mechanical strength, thermal conductivity and differential scanning calorimetry analysis.

## 1. Introduction

Shape memory materials have the property of remembering and returning to their original form in response to specific external stimuli such as heat, light, current and magnetic fields; heat is the main external stimulus [1,2,3,4,5,6,7]. Shape memory materials are transformed by applying an external force at a high temperature, and the shape is temporarily fixed when they are cooled. Later, they return to their original, permanent shape, which they remembered, at temperatures above the glass transition [8,9,10]. The shape memory effects of materials have been studied extensively since they were first discovered in Ni-Ti alloy at the U.S. Naval Ordnance Laboratory in 1963, and by the 1980s, they had been put into practical use. Since then, the Nippon Zeon Company developed the first shape memory polymer polynorbornene in the early 1980s, the second discovery, a trans-isopolypreme-based feature shape memory polymer, was developed in Kuray, and later a shape-memory polymer based on styrene butadiene, was developed by Asahi [11,12].

SMPs have the following property: if they attain a rubbery state at a temperature above the glass transition temperature, the elastic modulus lowers rapidly. Subsequently, when the surrounding temperature cools down, they display a temporary shape. If the surrounding temperature rises above the glass transition temperature again, they return to the glassy state and the elastic modulus increases, thereby resulting in the original shape [13]. Recently, the development of composite materials that exploit these characteristics of shape-memory materials, has attracted much attention. Shape memory alloys (SMAs) have advantages such as biocompatibility and a bidirectional shape memory capability, but they also have disadvantages such as limited rigidity and processing conditions. Shape memory polymers (SMPs) have a lower density than shape memory alloys and can thus be easily realized with lightweight characteristics. They also have excellent shape memory characteristics and excellent biocompatibility. In addition, SMPs can be applied in various fields due to their low price and good formability; they have attracted attention for use in wearable displays and biomedical materials due to their excellent biocompatibility. Nevertheless, because of their polymer properties, pure SMPs have inferior mechanical properties relative to SMAs, and the use of SMPs in a wider range of engineering applications has remained limited. Studies to reinforce the physical properties, such as the tensile strength of SMPs have thus been carried out. At present, most of the research on shape memory materials are mainly conducted in academia and on metal-based alloys; moreover, it is in the early stages of research.

In this study, we have attempted to find optimized curing conditions of shape memory polymer composites and have also analyzed shape memory performance and their physical properties according to the types of fillers and hardener amount. In addition, by adding graphite, we investigated the mechanical properties of the composites with improved shape memory performance.

## 2. Experimental Procedure

### 2.1. Materials and Preparation

The SMP sample was prepared using an epoxy resin and hardener (Struers, Korea). The following reinforcements were added: graphite (Sigma Aldrich, USA), Cu (Yeeyoung Cerachem, Korea), Al (Yeeyoung Cerachem, Korea) and 60 μm and 100 μm carbon fiber (Fiberman, Korea). Figure 1 indicates the epoxy resin and hardener used in this work. The SMP sample preparation procedure is shown in Figure 2. First, the epoxy resin and curing agent were placed in a clean bottle in a 10:1 weight ratio and hand shaken until completely mixed. Subsequently, the various reinforcements were added to the mixture, which was mixed with a mechanical stirrer. The mixing ratios of the reinforcements are shown in Table 1. The SMP composite was cured in a mold of 100 mm × 20 mm × 1 mm for 2 h at 80 °C. The mold and the resulting samples are shown in Figure 2. The shape fixation rate and recovery rate tests were conducted on samples with ratios of epoxy resin to hardener of 8:1, 9:1 and 10:1, respectively. The shape fixation rate was the highest and the recovery time was the fastest for sample 3, for which the ratio of epoxy resin to hardener was 10:1. Therefore, the hardener ratio was fixed at 10:1, and the fillers were added. The shape recovery test was conducted with 40 wt% and 50 wt% fillers. However, because fractures occurred during deformation to the temporary shape, the maximum ratio of the additive was fixed at 30 wt%.

### 2.2. Methods

#### 2.2.1. Differential Scanning Calorimetry (DSC)

DSC thermal analysis of the shape memory polymer sample was performed on a DSC250 thermal analyzer (TA instrument, USA) to determine the glass transition temperature (T_g_) for the type and amount of the filler. The samples were heated from 30 °C to 180 °C in a protective atmosphere of N_2_ at a heating rate of 10 °C/min.

#### 2.2.2. Tensile Test

Tensile tests were carried out at room temperature using a universal testing machine (Instron 3382, USA) with the ASTM D638 standard. The gauge length was 50 mm, and the crosshead speed was 1 mm/min.

#### 2.2.3. Shape Memory Test

Samples of 100 mm × 20 mm × 1 mm were heated to 100 °C and then modified to fit a 20 mm thick mold by applying an appropriate force. The maximum bending angle recorded was *θ_max_*. The fixed mold is illustrated in Figure 3. The modified sample was cooled to room temperature under constant external force. The mold and force were removed after the sample was fixed, and the sample then kept at room temperature until it was fully fixed. The fixed bending angle is denoted as *θ_fixed_*. Finally, we maintained a constant temperature of 100 °C and recorded the bending angle (*θ_i_*), every 10 s. The process is illustrated in Figure 4.
(1)$$\mathrm{Shape}\;\mathrm{fixation}\;\mathrm{ratio}:\;\frac{\theta_{fixed}}{\theta_{max}}\times 100$$
(2)$$\mathrm{Shape}\;\mathrm{recovery}\;\mathrm{ratio}:\;\frac{\theta_{max}-\theta_{i}}{\theta_{max}}\times 100$$

#### 2.2.4. Thermal Conductivity

The thermal conductivity was measured using a Hot Disk (TPS 2500 S, Sweden) equipment to measure the change and improvement of the thermal conductivity according to the type and content of the filler.

## 3. Results and Discussion

### 3.1. Differential Scanning Calorimetry (DSC)

The results of DSC for the three ratios of epoxy resin to hardener are shown in Figure 5. The glass transition temperatures (T_g_) obtained from the DSC analysis are summarized in Table 2. Polymer segments are assumed to be locked into a glassy state when the polymer segmental motion is limited at temperatures below T_g_. In this work, the midpoint of the temperature range of the DSC curve was defined as T_g_. The value of T_g_ was found to be between 59 °C and 73 °C, and the T_g_ values tended to decrease with a relative decrease in the amount of hardener. The results of DSC according to the filler type and amount are shown in Figure 6, and the T_g_ values obtained from the DSC analysis are summarized in Table 3. This result shows that all samples with Graphite and Cu fillers, have a distinctive T_g,_ ranging from 46.41 °C to 58.37 °C and their T_g_ value decreases as the amount of filler increases. It is expected that the overall transition temperature can be tuned by adding fillers. The fillers make the thermal conductivity of SMP composites relatively higher. The typical segments of these SMP composite added fillers would respond faster than SMPs without filler for the same amount of heat. Thus, the transition temperature of composites shift to a lower value by adding filler [14].

### 3.2. Tensile Strength Test

The tensile strength according to the ratio of the epoxy resin to the hardener and the type of filler is shown in Figure 7. The tensile strength of the specimen with a curing agent to resin ratio of 8/1 was 127% higher than that of the 10/1 specimen. As the ratio of the curing agent increased, the mechanical strength improved, but the shape-recovery ability tended to decrease. As the hardening agent ratio increases, the tensile strength of composites also increases, which is due to the larger number of 3D cross-linking networks during the curing process of epoxy. Generally, because fillers play important roles in enhancing mechanical properties, the tensile strength of composites may also vary according to the amount and shape of fillers. Generally, by using nanofillers such as graphite, copper and carbon fiber, the tensile strength of composites can be improved by adding appropriate amounts of fillers. However, the reported mechanical properties do not reflect the expected level of improvement, which can be attributed to the poor dispersion effect of the filler, agglomerates that act as crack initiation and weak interfacial interactions.

### 3.3. Shape Memory Test and Behavior

#### 3.3.1. Shape Fixation Ratio

The fixation rate tended to decrease as the additive content increased, but the fixation rate was maintained above 90%. A shape fixation ratio graph according to type and amount of filler is shown in Figure 8 and Table 4. Generally, the use of Cu as fillers yield a high fixation rate. When carbon-based filler graphite flake of average 20 µm size was used, the fixation rate tended to be substantially lower. For Cu, the 1 µm diameter particles were spherical and did not affect the fixation rate, whereas the plate-like graphite was considered to affect the shape fixation rate. SEM images of the fillers are shown in Figure 9.

#### 3.3.2. Evaluation of Shape Recovery Rate

The shape recovery ability was tested according to the filler type and content. Eventually every samples with filler show the permanent shape is fully recovered regardless of time. Thus, the comparative analysis by type filler was conducted on the capability of shape recovery over a specific period of time. A graph of the shape recovery ratio versus time, is shown in Figure 10. The recovery rate from the tests are shown in Table 5 and Table 6. Samples reinforced with nanocarbon materials such as graphite tended to have an increased recovery rate as the filler amount increased, whereas in the samples containing Cu, the recovery rate decreased as the filler amount increased. Figure 11 shows snap shot images of measuring the shape recovery ratio according to the type and filler content.

### 3.4. Thermal Conductivity of SMP Composites

The thermal conductivity was measured to investigate the changes in thermal conductivity according to the filler type and content, and to examine the correlation between the thermal conductivity and shape recovery capability. A graph of the thermal conductivity according to the filler type and amount, is shown in Figure 12. For comparison, a pure SMP sample without additives was prepared and measured: it exhibited a low thermal conductivity of 0.2296 W/mK.

By adding only 10 wt% of graphite, the thermal conductivity was doubled. In addition, it was also observed that, the higher the amount of fillers, the higher the thermal conductivity. The highest thermal conductivity was 0.898 W/mK for 30 wt% graphite composites. However, for the copper filler, the effect of filler amounts was not as significant. The highest thermal conductivity for copper filler was 0.2776 W/mK for 10 wt% amount. For this reason, graphite, which has higher thermal conductivity, shows better shape recovery performance. This indicates that graphite is a more effective filler than Cu for SMPs in terms of thermal conductivity and the thermal reaction of shape recovery. This is because the volume of graphite filler is much larger than that of copper for the same weight, so the dispersibility of graphite in SMP composites may be higher than that of copper. Figure 13 shows the volume comparison of graphite and copper for the same amount.

## 4. Conclusions

SMPs have poor shape recovery performance such as rate and time to full recovery for without fillers, so their applicability is low. The thermal conductivity plays an essential role in the improvement of shape recovery capability. Therefore, in this study, commonly available fillers with the highest thermal conductivity were considered to enhance their applicability and supplement their thermal conductivity and shape recovery performance. DSC results were also provided. On comparing the shape recovery ability according to the filler type under the same conditions, the specimens with graphite additives exhibited a tendency of increased thermal conductivity, as the filler content increased. For the specimens with non-carbon-based Cu additives, the higher the filler content was, the lower was the directional recovery ability. Nevertheless, for all specimens, the final recovery rate exceeded 90%.

## Figures and Tables

**Figure 1 micromachines-12-01107-f001:**
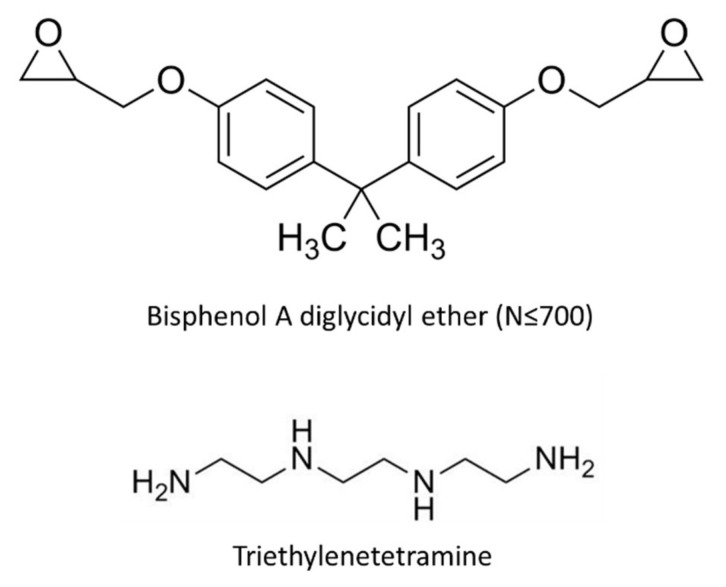
Materials used to synthesize shape memory polymer.

**Figure 2 micromachines-12-01107-f002:**
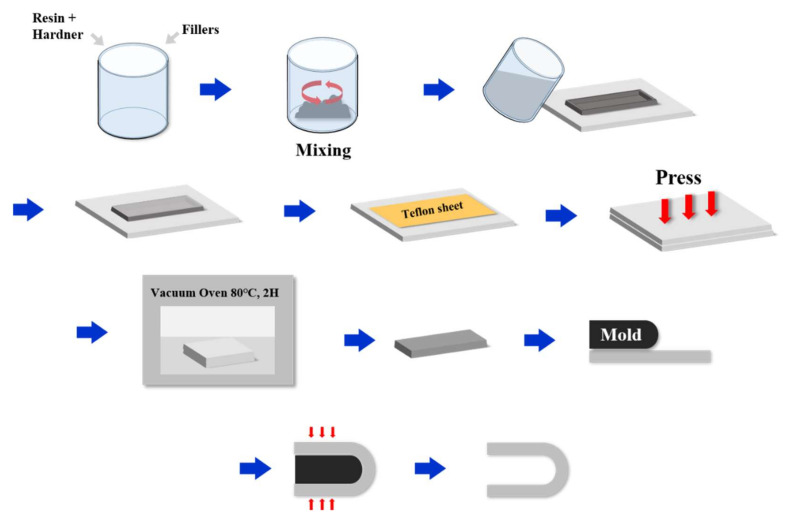
Process for the Test of Shape Memory Sample Preparation.

**Figure 3 micromachines-12-01107-f003:**
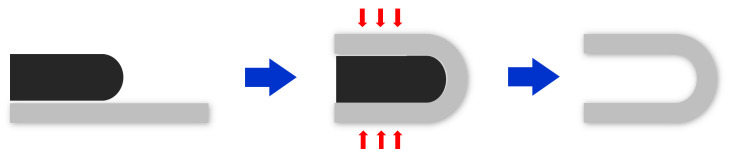
Process for the Test of Shape Fixation Properties.

**Figure 4 micromachines-12-01107-f004:**
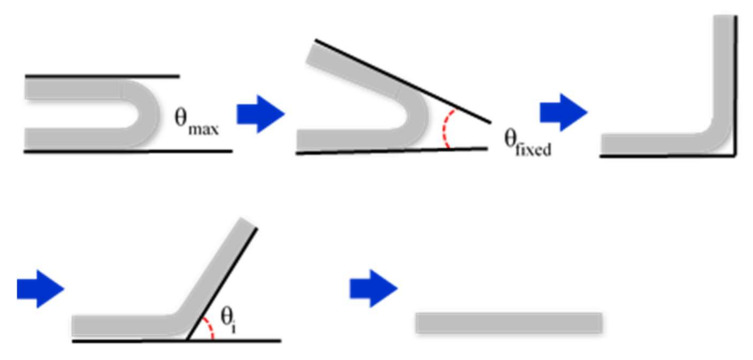
Process for testing the shape fixation properties: maximum bending angle and fixed bending angle.

**Figure 5 micromachines-12-01107-f005:**
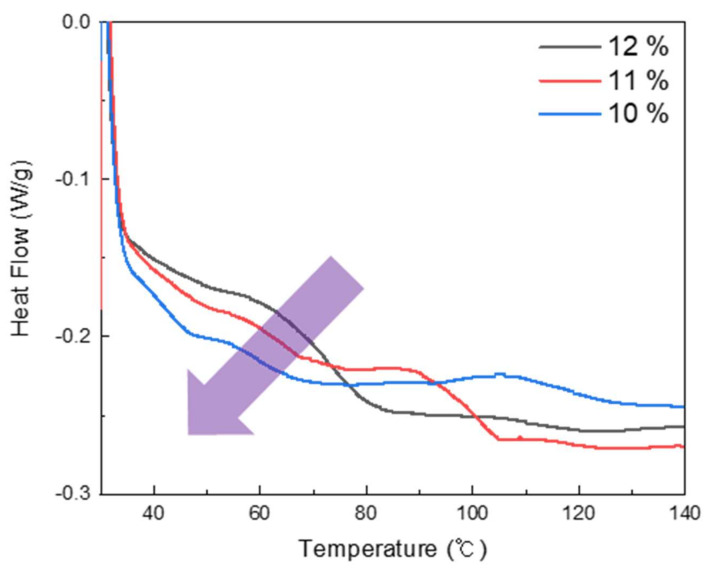
Results of DSC data for three ratios of hardener to epoxy resin.

**Figure 6 micromachines-12-01107-f006:**
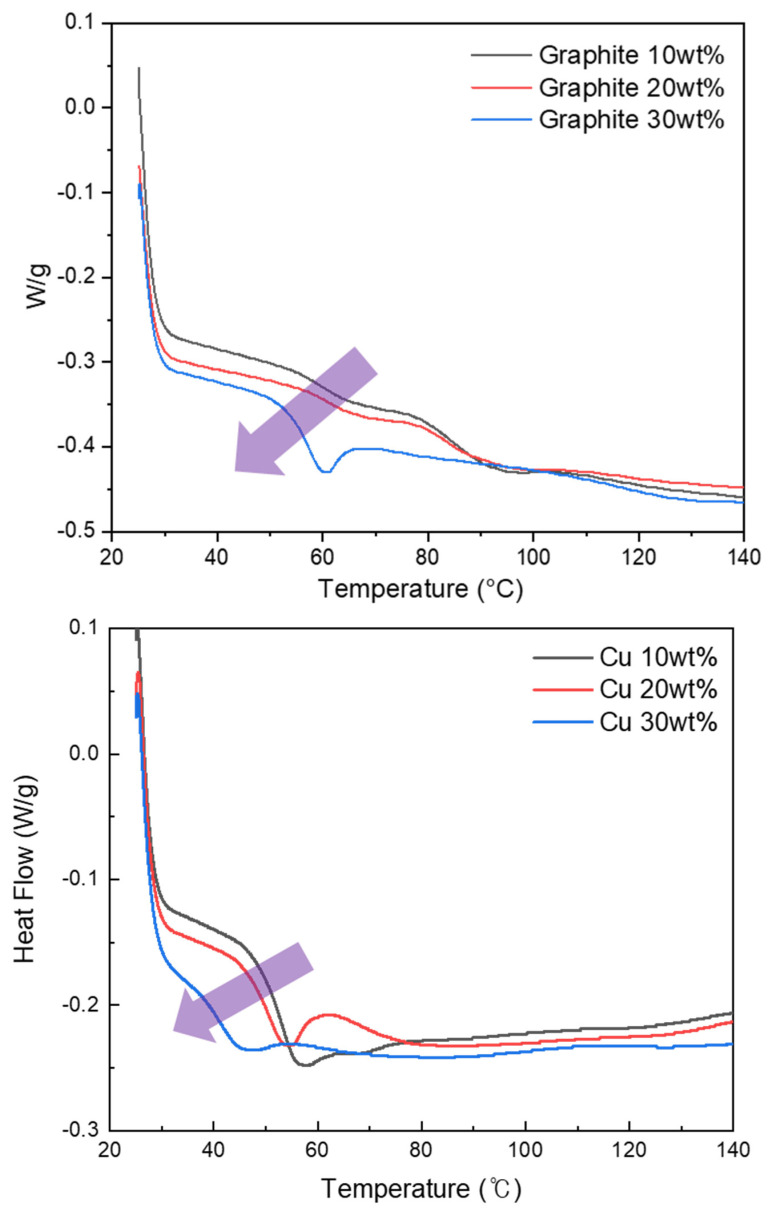
Results of DSC according to filler type and amount with Epoxy 10/1.

**Figure 7 micromachines-12-01107-f007:**
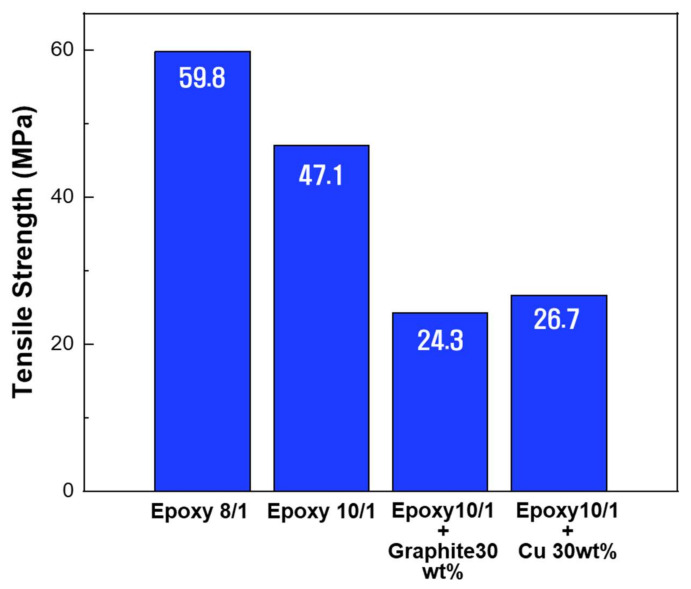
Results of Tensile strength for three ratios of epoxy resin to hardener.

**Figure 8 micromachines-12-01107-f008:**
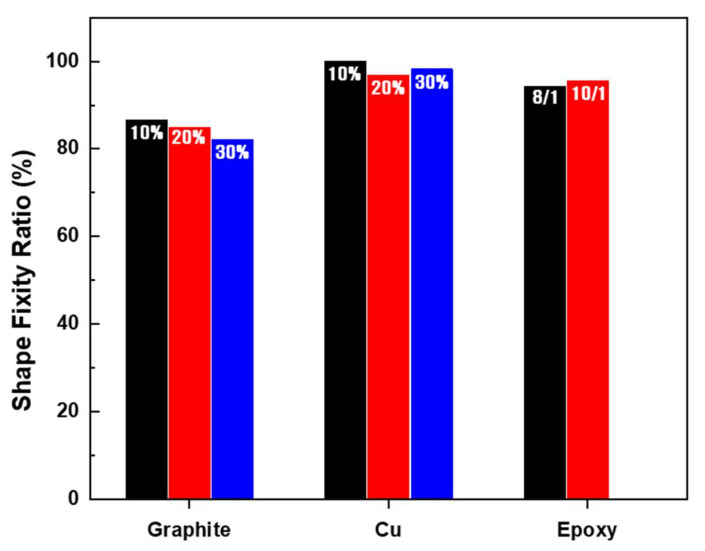
Comparison of the shape fixation ratio by amount and type of filler.

**Figure 9 micromachines-12-01107-f009:**
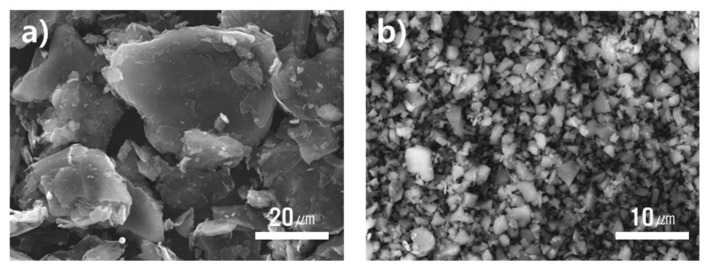
SEM images of the fillers. (**a**) Graphite (Avg. 20 μm), (**b**) Cu (Avg. 1 μm).

**Figure 10 micromachines-12-01107-f010:**
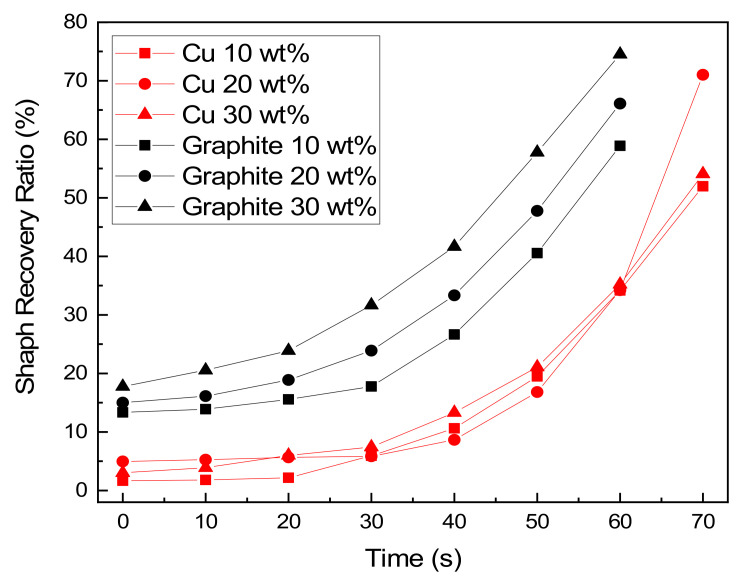
Shape recovery rate by type of filler and amount.

**Figure 11 micromachines-12-01107-f011:**
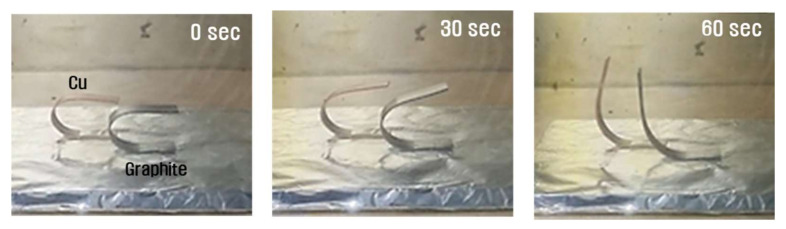
Snap shots of shape recovery performance by type of filler.

**Figure 12 micromachines-12-01107-f012:**
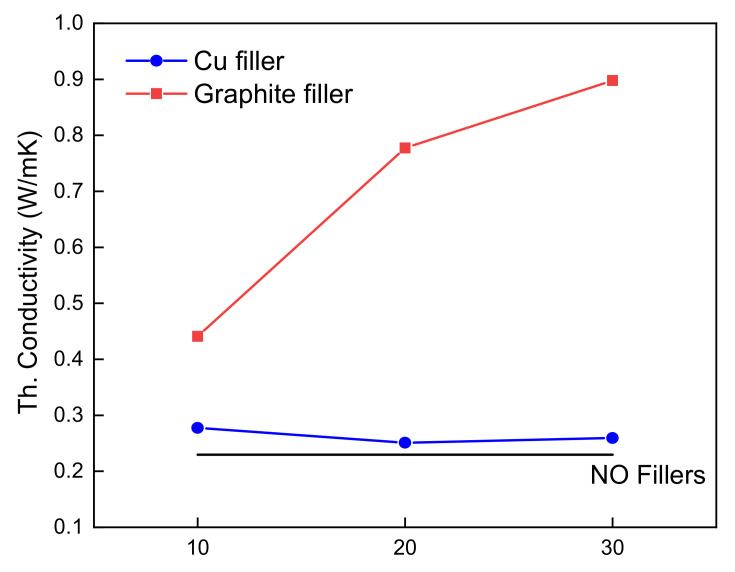
Thermal conductivity graph by amount and type of fillers.

**Figure 13 micromachines-12-01107-f013:**
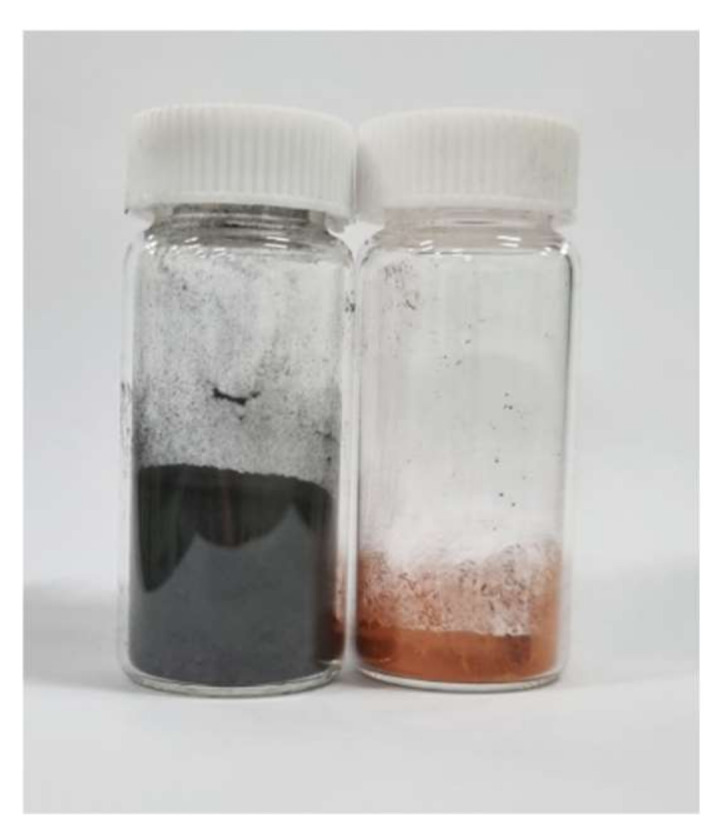
Image of the same weight of graphite and copper volumes (3 g).

**Table 1 micromachines-12-01107-t001:** Composite mixing ratios.

No.	Epoxy/Hardener	Graphite Amount	Cu Amount
1	8/1 (12%)	-	-
2	9/1 (11%)	-	-
3	10/1 (10%)	-	-
4	10/1	10 wt%	-
5	10/1	20 wt%	-
6	10/1	30 wt%	-
7	10/1	-	10 wt%
8	10/1	-	20 wt%
9	10/1	-	30 wt%

**Table 2 micromachines-12-01107-t002:** Glass transition temperatures for three ratios of hardener to epoxy resin.

Hardener Amount	10%	11%	12%
T_g_ (°C)	72.04	62.87	59.91

**Table 3 micromachines-12-01107-t003:** Glass transition temperatures for the amount and type of fillers.

Sample	Graphite			Cu		
	10 wt%	20 wt%	30 wt%	10 wt%	20 wt%	30 wt%
T_g_ (°C)	55.50	58.37	53.51	56.88	54.10	46.41

**Table 4 micromachines-12-01107-t004:** Shape fixation ratio.

Shape Fixation Ratio	Epoxy		Graphite			Cu		
	8/1	10/1	10 wt%	20 wt%	30 wt%	10 wt%	20 wt%	30 wt%
Rate (%)	94.30	95.52	86.66	85.0	82.22	100	96.97	98.33

**Table 5 micromachines-12-01107-t005:** Shape recovery rate at 60 s of experiment.

Shape Recovery Ratio	Epoxy		Graphite			Cu		
	8/1	10/1	10 wt%	20 wt%	30 wt%	10 wt%	20 wt%	30 wt%
Rate (%)	27.75	55.66	58.88	66.11	74.52	34.17	34.22	35.22

**Table 6 micromachines-12-01107-t006:** Time when the shape recovery rate is 50%.

Shape Recovery Ratio	Epoxy		Graphite			Cu		
	8/1	10/1	10 wt%	20 wt%	30 wt%	10 wt%	20 wt%	30 wt%
Time (sec)	76	57	55	51	45	69	64	68

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
