# Peer review of "Analysis of Shape Memory Behavior and Mechanical Properties of Shape Memory Polymer Composites Using Thermal Conductive Fillers"

_micromachines, 2021, doi:10.3390/mi12091107_

Round 1

Reviewer 1 Report

This reviewer has made a careful assessment of this manuscript, which investigate the shape memory performance and mechanical properties of Shape Memory Polymer Composites. this manuscript is very interesting and generally well written. There are no corrections to require, therefore this reviewer suggests this manuscript to be accepted in its present form.

Author Response

Response to Reviewr 1:

Thanks for your careful reading and comments.
I look forward to hearing from you in due time regarding our submission and to respond to any further questions and comments you may have. 

Reviewer 2 Report

The manuscript was devoted to finding optimized shape memory polymer composites and investigating the shape memory performance and physical properties according to the filler type and amount of hardener. The manuscript addresses a novel and efficient topic which has recently been boomed. Moreover, it is well-written and the purposes, advantages and the results of this work are reasonably highlighted. Thus, the reviewer recommends publishing of the manuscript after addressing the following comments.

  1. What is the exact carbon nanomaterial? It is CNT (Carbon nanotube)? What are the aspect ratios of the carbon nanomaterials?
  2. What is the volume fraction (weight percentage) of carbon nanomaterial? Is it 10 wt%, 20 wt%? it seems high weight percentage of nanomaterials which may cause agglomeration in processing of final composite material. How are authors confident that the nanomaterial is fully/uniformly dispersed without any agglomeration?
  3. Regarding Fig. 7, increasing the carbon nanomaterial weight percentage in polymer, leads to increase in tensile strength according to many published articles. While in Fig.7, an inverse result is obtained; How do authors justify this issue? Isn’t it essential to carry out the experiment again?
  4. Based on the shapes and sizes of the graphite flakes (which their sizes are bigger than Cu and their shapes are plate-like), the authors should discuss that why the use of Cu fillers will yield a high fixation rate.

Author Response

Please see attached file for the response to the review 2 comments.
